# Relationship between Food Habits, Nutritional Status, and Hormone Therapy among Transgender Adults: A Systematic Review

**DOI:** 10.3390/nu16193280

**Published:** 2024-09-27

**Authors:** Ivo P. Sousa, Teresa F. Amaral

**Affiliations:** 1FCNAUP, Faculty of Nutrition and Food Sciences, University of Porto, 4150-180 Porto, Portugal; ivopaulosousa@gmail.com; 2Municipality of Vila Nova de Gaia, Health Division, 4430-999 Vila Nova de Gaia, Portugal; 3LAETA-INEGI/FEUP, Associated Laboratory of Energy, Transports and Aerospace, Institute of Science and Innovation in Mechanical and Industrial Engineering, Faculty of Engineering, University of Porto, 4200-465 Porto, Portugal

**Keywords:** transgender, hormone therapy, public health, nutritional status, eating behavior, systematic review

## Abstract

**Background/Objectives**: The current gender-specific nutritional assessment methods for the transgender population may not cover the unique physiological characteristics of the gender transition process. Considering the potential effects of hormone therapy (HT), it has become relevant to review current evidence on the nutritional status of the transgender population. This systematic review aims to provide an updated report of the characteristics of the nutritional status, including food habits, and eating disorders in transgender individuals undergoing HT. **Methods**: Five databases were researched (PubMed, Web of Science, Scopus, Scielo, and Cochrane Library) from database inception to May 2024. The PRISMA 2020 statement was used. Studies focusing on adult transgender individuals (18 to 65 years old) that included outcomes related to nutritional status, HT, and food habits were considered for this review. The NOS and NIH tools were chosen to perform the risk of bias and quality assessment. **Results**: A total of 122 studies were identified, and 27 were included in this review. These studies comprised sixteen cohorts, seven cross-sectional, and four case studies, with a combined number of 8827 participants. BMI was the most referenced parameter, varying between low weight and overweight. High food insecurity frequency, restricted eating behaviors, high fat intake, and low levels of vegetable, grain, and fruit consumption were also observed. **Conclusions**: While nutritional status was perceived as a relevant factor when administering HT, the relationship between HT with both nutritional status and food habits has been insufficiently explored and warrants further research.

## 1. Introduction

The transgender population has specific nutritional concerns related to the potential effects of gender transition procedures [1,2,3]. In this population, the condition “gender dysphoria” (GD) is observed in some cases. It is defined as a “deep and persistent conviction that gender identity (self-identification as female or male) is not in accordance with physical appearance and/or anatomy” [4]. To achieve the goal of adjusting their physical body to their gender identity, transgender individuals may resort to the use of hormone replacement therapy (HT) to alter physical attributes depending on the gender identity in question. Trans women (TW) may use estrogen for feminizing transitions, while trans men (TM) may resort to testosterone for masculinizing transition purposes [1,2,3]. Other individuals who identify as non-binary (NB) or gender nonconforming (GNC) may or may not seek out HT, depending on their desire to transition [1,2,3]. It is important to note that a transgender individual may not be going through GD or going through a physical transition.

The effects of HT include weight gain, eating disorders, altered lipid profiles, and cardiovascular risk [1,2,3]. Due to these alterations, the current recommendations state that the nutrition guidelines for one’s biological sex should be used to meet energy and nutrient needs if HT has not started [5]. However, no guidelines regarding nutritional care in the transgender population exist, and the conditions that influence the nutrients and energy needs of this target population have yet to be determined [5,6,7,8]. The insufficient level of knowledge and skills about transgender health among general health and nutrition professionals has also been reported as low [9,10].

The transgender population is at increased risk of specific health and nutritional outcomes due to body dissatisfaction, weight complications, significant health care disparities, food insecurity, experiences of discrimination and stigmatization, and adverse effects derived from HT [6,7,8,9,10,11]. The effects of HT related to the nutritional status of transgender individuals taking HT have been researched [11,12,13,14]. Given the wide variety of HT regimens, usage periods, exposure, and potential confounding factors like the duration of usage, this association has been complex to assess [6]. A meta-analysis by Klaver et al. was conducted to investigate changes in body weight, body fat, and lean body mass during HT [6]. In feminizing transitions, weight changes reflected increased fat mass and decreased lean mass due to the administered hormones [6]. In masculinizing transitions, an increase in lean body mass and a decrease in fat mass was observed [6].

Seal et al. review of the literature focused on the impact of HT on cardiovascular diseases in trans and NB individuals [11]. For TW, myocardial infarction risk was lower in individuals taking HT than in non-trans females. For TM, there was an adverse effect of HT on the lipid profile, although the risk of cardiovascular disease was not above the general non-transgender male population [11].

Streed et al. reviewed the outcomes of HT administration on individual cardiovascular risk factors [12]. Specific causes for the increased risk of cardiovascular diseases were minority stressors, which influenced cardiovascular health behaviors and consequently increased the risk of poor mental and physical health outcomes, including nutritional status [12].

Within a scoping review concerning dietary intake, nutritional inequality, nutrition assessment methods, and HT in the transgender population, Rozga et al. found that ten out of the eighty-nine studies were focused on dietary intake [13]. Of those ten studies, three were centered exclusively on alcohol intake and mental health concerns, and none focused on dietary patterns, eating behaviors, or micronutrient intake, three essential components in nutrition research, emphasizing the need to increase knowledge in this field [13].

Lastly, there are studies describing how nutrition relates to specific gender experiences in the transgender population [14,15]. A systematic review by Gomes et al. reported body image and weight control, food and nutrition security, nutritional status, nutritional health assistance, and the perception of healthy eating [14]. According to this review, there was a high prevalence of food insecurity, ultra-processed food habits, and compromised access to nutritional assistance [14]. In addition, a recent study also conducted by Gomes et al. [15] further highlighted that the trans community is susceptible to food insecurity, with 68.8% of their sample experiencing difficulty in accessing nutritionally adequate food sources [15].

More recently, Heiden-Rootes et al. published a scoping review investigating the literature on eating and body image for trans and NB adults [16]. Hormone therapy was found to be associated with a decrease in eating disorder (ED) symptoms and improved body image [16]. This becomes a crucial point given the higher prevalence of ED in transgender communities and is further corroborated by previous research conducted on this topic [17,18,19,20].

Nagata et al. developed community norms for the ED examination in transgender individuals [17]. It was revealed that in their sample, 8.1% of TW participants had a positive diagnosis of ED by mental health provider or physician [17]. Comparatively, their TM sample had a higher percentage (10.6%) of positive ED diagnosis [17]. This higher prevalence in TM compared to TW was also observed in Rasmussen et al.’s systematic review on ED symptomatology in transgender individuals [18]. According to this synthesis, both TM and TW had higher levels of ED symptomatology compared to non-trans individuals, with TM displaying higher levels compared to TW [18].

McGregor et al. examined and reviewed the available bibliography centered on the specific risk factors for transgender individuals with ED [19]. According to the review, due to extrinsic factors (minority stress; barriers to gender-affirming care), GD, and the desire to pass, there is an increased risk of ED in transgender individuals [19]. The desire to pass may extend to developing ED and related compensatory behaviors during the earlier stages of the gender transition process. These behaviors will be performed with the intent of masking the bodily features from the individual’s undesired sex at birth, as one of the most significant risks of ED in this population is body dissatisfaction [20,21]. In feminizing transitions, the user may attempt to suppress their larger body frame or muscle by practicing a restrictive diet, which may lead to compensatory behaviors such as the use of laxatives or purging [20]. In masculinizing transitions, weight loss may be seen as a method to suppress feminine features (breasts) and secondary sexual characteristics (menstruation), leading to an increase in ED prevalence [20]. Preparation for surgery may also be seen as a reason to undergo these restricted eating habits [20].

With these results in mind and given that research in this field is still in its early stages, with few published studies, it is opportune to systematize the available scientific knowledge on the impact of HT on nutritional status, ED, and food habits. In addition, no reviews have been carried out that focus exclusively on the association between HT, eating habits, nutritional status, and the characteristics of transgender individuals undergoing HT. This systematic review aimed to provide an updated report of the characteristics of the nutritional status, including food habits and ED in transgender individuals undergoing HT.

## 2. Materials and Methods

This systematic review was conducted using the PRISMA 2020 checklist as a guide [22]. The protocol was registered on the International Prospective Register of Systematic Reviews (PROSPERO) platform (CRD42022338551).

Five databases were researched: Pubmed, Web of Science, Scopus, Scielo, and the Cochrane Library. In addition, research was also undertaken via other methods, namely, websites and citation searches [22].

The expressions and keywords chosen for the literature search were separated into three groups. The first comprised different gender identities inside the trans umbrella and terms associated with it: “trans*”; “transgender”; “transsexual”; “transwoman”; “transman”; “gender nonconforming”; “non-binary”; “gender fluid”; “genderqueer”; “gender dysphoria”. The second included terms related to food habits and eating disorders: “food habits”; “food consumption”; “food intake”; “nutritional intake”; “nutrition”; “nutrition assessment”; “food choice”; “diet”; “eating behavior”. The third group was composed of HT and terms associated: “hormonal therapy”; “hormonal replacement therapy” and “gender-affirming therapy”.

All original studies centered on food habits, nutritional status, and eating disorders among the transgender population, from inception until May 2024, were included in this systematic review.

Other inclusion and exclusion criteria were established to carry out the screening procedures. Inclusion criteria comprised all articles that included information regarding food habits and trans/transgender individuals aged between 18 and 65 years, either undergoing HT or not. Articles that were not written in English or Portuguese, studies that included either solely underage participants (ages below 18 years old), solely elderly participants (ages 65 years and old), or participants with dementia or terminal illness were excluded. Review studies (which included other systematic reviews) and unpublished articles were also excluded.

The screening process was administered in three steps, following the criteria mentioned above to verify the eligibility of the articles. Firstly, articles were excluded based on the title and abstract. Secondly, after the initial title and abstract screening, the full texts of the remaining studies were analyzed and verified to determine the eligibility of the topic of the present systematic review. Finally, the previously mentioned texts were re-evaluated for their adequacy. Several indicators were considered for this review, including any health or nutritional indicator related to the food habits or nutritional status of transgender individuals undergoing HT and any health or nutritional indicator about the food habits of transgender individuals not undergoing HT.

Two reviewers were involved in the quality assessment. To assess the potential formal risk of bias, the Newcastle Ottawa Quality Assessment Scale (NOS) was used for cohort, case–control, and cross-sectional studies [23,24,25]. The following classification was used: 0–3 was seen as low quality, 4–6 as moderate quality, and 7–9 as high quality [23,24,25].

For case studies, the Study Quality Assessment Tool developed by the National Heart, Lung, And Blood Institute (NIH) alongside its respective criteria (fair, good, and poor, depending on the results) was applied [26].

The analysis was conducted according to the assessment of the food habits and nutritional status of the transgender participants undergoing HT. Every mention of food–eating-related behavior and nutritional status-related measures or variables was included in this part of the analysis. A study would be included in this review if it could either answer the proposed objectives, despite having no data to answer other objectives or include relevant outcomes that could be used to answer the proposed objectives.

Two reviewers applied the eligibility criteria and selected the studies for inclusion in this systematic review, and the other confirmed the process. In case of disagreement, the reviewers discussed the subject based on the support of the literature to find coherence.

Concerning data extraction, the following information was collected: name of the first author; year of publication; country; study design; sample size; setting; nutritional status; food habits or food intake; eating disorders; age; gender identity; and HT. The methodology used to assess nutritional status, food habits, eating disorders, and results regarding those parameters were also extracted. Identified records were de-duplicated using systematic review assistant deduplication followed by a manual search in Endnote.

## 3. Results

Twenty-seven studies were selected for this review with a combined number of 8827 participants, with 3557 confirmed to be under the trans umbrella [27,28,29,30,31,32,33,34,35,36,37,38,39,40,41,42,43,44,45,46,47,48,49,50,51,52,53]. A PRISMA 2020 flow diagram of the literature search can be seen in Figure 1. All twenty-seven studies are summarized in Table 1.

The gender identities included in the articles were TW (*n* = 1802) [29,31,33,35,36,38,39,41,42,44,46,47,48,49,51,52], TM (*n* = 1202) [27,28,29,30,31,32,33,36,37,39,40,42,43,45,46,48,49,50,51,52,53], and GNC (*n* = 64) [27,39]. The remaining participants’ gender identities were either not specified or were grouped into a broad term, such as “transgender individuals” [34] or “written-in different identity” [39]. The studies screened included sixteen cohort studies [28,29,30,31,32,33,36,37,43,44,45,48,49,50,51,52], seven cross-sectional studies [27,34,38,39,42,47,53], and four case studies [35,40,41,46]. The oldest study was from 1998, and the most recent one was published in 2024. Regarding the geographical area, fifteen studies were from Europe, eleven were from America, and one was from Asia.

Concerning the quality scores (Table 2, Table 3 and Table 4), cohort studies had a mean NOS score of 6.3 ± 0.7 (minimum: 5; maximum: 7), and the cross-sectional studies had 5.6 ± 0.9 (minimum: 4; maximum: 7). Case studies ranged from fair to good, according to the NIH tool. The representativeness of the exposed cohort/sample was where the articles screened lost points in the NOS score. Some studies had a small community sample, such as the Kirby and Linde cross-sectional survey with 26 college students [39] or the Sánchez Amador et al. comparative survey with 37 TW in 105 participants [47]. Regarding the case studies, only one, performed by Linsenmeyer et al. [40], had an extensive description of the statistical methods used in the analysis and the results obtained. Even with these points, all studies screened provided a good groundwork to summarize the characteristics of the nutritional status for this review.
nutrients-16-03280-t001_Table 1Table 1Characteristics of the 27 studies included in this systematic review, organized by study type and number of participants.First Author, Year of PublicationCountryStudy DesignSample Size (*n*)and SettingNutritional StatusFood Habits/IntakeEating Disorders/MethodAgeGender IdentityHormonal TherapyOutcome Related ResultsEwan et al. (2014) [35]United States of AmericaCase reportOne reportEmergency centerWeight; body mass index; estimated mean body weightNot assessedAnorexia nervosa, assessed during a hospital visit (no assessment method specified)19 yearsTrans womanGonadotropin-releasing hormone agonist therapy and surgery, although the patient had not begun either treatment at the time of the report.Nutritional Status:Weight loss of around 36 kg, from 75 kg to 39 kg, during one year. From a body mass index of 26.8 kg/m^2^ to 13.8 kg/m^2^. Evidence of dehydration.Eating Disorders:Restricted energy intake, excessive use of laxatives, diet drugs, including weight loss supplements. The participant also showed compensatory food behavior, such as bingeing and purging via vomiting.Maheshwari et al. (2021) [41]United States of AmericaCase studyTwo casesSpecialized clinicBody mass index; physical examNot assessedHistory of undernutrition due to restrictive food intake disorder.Assessed via medical history and physical examination.28 and 24 yearsTrans womenSpironolactone and estradiol; spironolactone and leuprolideNutritional Status:Body mass index (only assessed in Case 1) = 17.2 kg/m^2^.Eating Disorders:One of the cases had a history of undernutrition due to avoidant/restrictive food intake disorder. Before initiating the low-dose spironolactone treatment, the nutritional concerns were addressed.Linsenmeyer et al. (2020) [40]United States of AmericaCase study10 casesGeneral trans men populationBody mass index; body fat percentage; estimated energy requirements; waist circumferenceDietary pattern analysis was conducted using a three-day food diary and ESHA Food Processor Nutrition Analysis software. (available at https://esha.com/products/food-processor/, accessed on 18 May 2024)Dietary Guidelines for Americans 2015–2020wasused as reference.Degree of eating competence and the risk of eating disorders, assessed with the EAT-26 and ecSI-2 tools.Higher than 18 yearsTrans menGender-affirming medical interventions: hormone therapy or surgeries.Nutritional Status:70% of the sample was obese, both according to the body mass index (33.8 ± 9.3 kg/m^2^) and body fat % (average 31%); 30% were obese class I, 20% obese class II, and 20% obese class III; 60% had a high waist circumference (42″ ± 8.8). Two cases, however, had their waist circumferences and body fat percentage within a healthy percentage despite an overweight classification.Food Habits/Intake:Energy intake ranged from 58 to 110% of the estimated needs. The sample did not meet the recommendations for fiber or fruit and vegetable intake. A diet high or marginally high in sodium and saturated fat was observed, low or marginally low in potassium and vitamin D, and high in calcium and iron.Eating Disorders:None of the ten participants screened positive for disordered eating according to the EAT-26 tool (average score 6.4 ± 4.6.According to the ecSI-2.0, two of the patients scored a high degree of eating competence (average Score = 28.4 ± 8.6)Resmini et al. (2008) [46]ItalyCase study26 cases(15 trans women; 11 trans men).Body mass indexNot assessedNot assessedTrans women = 33.2 ± 2.1 yearsTrans men = 30.9 ± 1.8 yearsTrans women; trans menEstradiol; anti-androgen; testosterone; estrogenNutritional Status:The sample consisted of subjects without dyslipidemia and with normal body mass index. Trans women = 21.4 ± 0.62 kg/m^2^; trans men = 21.45 ± 0.57 kg/m^2^.Kirby and Linde (2020) [39]United States of AmericaCross-sectional26 college students from Public Midwestern university(24 trans students: 1 trans woman; 7 trans men; 6 gender nonconforming; 10 written-in different identities)Weight, height, andbody mass index, derived from the University’s College Student Health Survey.Nutrition knowledge and skills; dietary intake during seven days derived from the University’s College Student Health Survey.Weight loss attempts, weight loss methods, and binge eating questions derived from the University’s College Student Health Survey. Food insecurity derived from the Rainbow Health Initiative’s Voices of Health.Mean = 22.7 yearsTrans women;trans men;gender nonconformingGender-affirming medical interventions: hormone replacement therapy and gender-affirming surgery27% of the participants were on hormone replacement therapy.Nutritional Status:Average body mass index = 24.9 kg/m^2^.Food Habits/Intake:During the seven days, 46% did not eat fruit daily; 42% did not eat vegetables daily; 58% of participants did not eat whole grain food products; over 50% of participants reported eating less due to not having resources, including over a third who reduced the size of meals and skipped meals or went hungry.Eating Disorders:Over a third of participants followed a restricted diet to lose weight; 31% engaged in binge eating over the past 12 months; 50% attempted to lose weight; and 88% changed their eating or exercise behaviors to change their body.Yaish et al. (2021) [53]IsraelCross-sectional study56 participantsTransgender health centerBody mass indexNot assessedNot assessedMean = 25.9 yearsTrans menTestosteroneNutritional Status:Baseline body mass index:24.05 (21.3–31.2) kg/m^2^Body mass index of the no polycystic ovary syndrome/testosterone treatment group:23.0 (20.95–26.75) kg/m^2^Body mass index of the polycystic ovary syndrome/testosterone treatment group:26.7 (21.8–32.4) kg/m^2^Without differences between the two groups (*p* = 0.1).Sánchez Amador et al. (2024) [47]SpainCross-sectional comparative study105 participants (37 trans women participants)Anthropometrics (weight, height, and body mass index)Body composition via Bioelectrical impedance analysis (fat mass; muscle mass; fat-free mass)Not assessedNot assessedTrans women (mean 28.6 years, range 20 to 38)Trans womenOral estradiol valerate and oral cyproterone acetate for more than six monthsNutrition Status:The trans women sample had a higher body mass index (25.7 ± 4.1 kg/m^2^) than both non-trans men (22.9 ± 2.5 kg/m^2^) and non-trans women (22.0 ± 2.5 kg/m^2^) comparative groups. Similarly, the fat mass was 41% higher in the trans women sample compared to the non-trans men.Regarding fat-free mass, it was lower in trans women when compared to the non-trans men sample, with no significant differences.When compared with the non-trans women sample, the fat-free mass was higher in the trans women group by 24%.Arikawa et al. (2021) [27]United States of AmericaCross-sectional analysis of a questionnaire239 participants who identified as being LGBTQ+(59 trans male; 58 gender nonconforming)Non-probability volunteer sampleBody mass index; weight changesNot assessedFood security score; Eating attitudes test score, eating disorder examination self-report questionnaire score18 to 35 yearsTrans male;gender nonconformingSex-steroid hormonesNutritional Status:Body mass index average (kg/m^2^): trans men: 27.4 (25.4–29.4); gender nonconforming: 27.4 (25.4–29.4).66% of trans men participants and 67% of gender nonconforming participants underwent weight change during the past year.Eating Disorders:54.4% reported food insecurity; 31.4% of study respondents exhibited eating disorder pathology; 28% of study participants reported that they engaged in eating disorder behaviors; the eating disorder behavior most frequently reported by respondents was binge eating.Diemer et al. (2015) [34]United States of AmericaCross-sectional479 participantsNot assessedNot assessedAmerican College Health Association questionnaire(past year eating disorder diagnosis; past month diet pill use; past month vomiting or laxative use)Median = 20 yearsTransgender individualsNot assessedEating Disorders:15.82% of the transgender sample was diagnosed with an eating disorder in the past year at the time of the study.Self-reported eating disorder diagnosis and past month use of diet pills and vomiting or laxative was higher among transgender students, who had higher odds of past year eating disorder diagnosis (odds ratio: 4.62, [3.41–6.26]), past month diet pill use (odds ratio: 2.05, [1.48–2.83]), and past month vomiting or laxative use (odds ratio: 2.46, [1.83–3.30]) compared to non-trans women.Martinson et al. (2020) [42]United States of AmericaCross-sectional1457 participants(1085 trans women; 372 trans men)Center for transgender medicine and surgeryBody mass index (recorded at the initial surgical consult, and the most recent subsequent visit); self-monitored weight managementNot assessedNot assessedMean = 35.4 ± 11.6 yearsTrans man/transmasculine spectrumTrans woman/transfeminine spectrum Not assessedNutritional Status:26% of the participants were obese (body mass index higher than 30 kg/m^2^) at the initial surgical consult, and 32% were overweight (body mass index between 25–29.9 kg/m^2^).No changes were noticed in the rate of obesity among trans and non-binary participants despite self-monitored weight management.Hojbjerg et al. (2021) [38]Denmark, Norway, Sweden, Finland, IcelandCross-sectionalquestionnaire survey11 specialized clinics with 4838 affiliated patientsBody mass indexNot assessedNot assessedNot applicableTrans womenClinical practice of feminizing hormone therapy; estradiol; testosteroneNutritional Status:From the list of risk factors used by clinics influencing the choice and dosage of hormonal treatment, BMI was used in ten of the twelve participating clinics. One clinic mentioned severe obesity, and another mentioned diabetes as another risk factor.Berra et al. (2006) [28]ItalyProspectivestudy16 participantsHospital SettingAnthropometrics (body weight; body mass index; waist circumference, body lean mass, and fat mass)Not assessedNot assessedMean:30.4 ± 5.4 yearsTrans menTestovironNutrition Status:Significant changes were observed in the body mass index and waist circumference after 6 months of testoviron treatment (*p* < 0.001, *p* < 0.001 and *p* < 0.05, respectively).Weight (kg): 58.2 ± 8.7 [at the start] vs. 60.4 ± 7.2 [after 12 months].Body mass index (kg/m^2^): 21.8 ± 2.9 [at the start] vs. 22.8 ± 2.6 [after 12 months].Waist circumference (cm): 77.1 ± 9.4 [baseline] vs. 78.1 ± 17.6 [after 12 months].Fat mass (kg): 16.5 ± 9.0 [at the start] vs. 14.2 ± 7.2 [after 12 months].Fat mass (%):27.1 ± 10.7 [at the start] vs. 22.4 ± 9.4 [after 12 months].Lean mass (kg): 40.9 ± 5.2 [at the start] vs. 46.9 ± 4.7 [after 12 months].Lean mass (%):72.3 ± 11.6 [at the start] vs. 77.6 ± 9.4 [after 12 months].Van Caenegem et al. (2015) [50]BelgiumProspective controlled study as a part of a large prospective study (ENIGI) Clinical Study23 participantsSexology and gender problems centerAnthropometrics (weight; height; body mass index; waist and hip circumferences; total body–fat mass)Not assessedNot assessedMean: 27 ± 9.0 yearsTrans menTestosterone undecanoateNutritional Status:After one year, hormone therapy increased lean body mass (10.4%) and decreased total body fat (9.4%). Unchanged BMI and waist and hip circumference were reported, however.Body mass index (kg/m^2^): 24.5 ± 5.3 (at the start) vs. 25.2 ± 4.1 (12 months).Waist circumference (cm): 78.4 ± 14.2 (at the start) vs. 78.2 ± 11.3 (12 months).Hip circumference (cm): 99.1 ± 7.9 (at the start); 98.4 ± 7.4 (12 months).Waist-to-hip ratio: 0.8 ± 0.1 (at the start) vs. 0.8 ± 0.1 (12 months).Total body–fat mass (kg): 19.9 ± 8.7 (at the start) vs. 17.5 ± 6.7 (12 months).Total body–fat mass (%): 29 ± 7 (at the start) vs. 25 ± 6 (12 months).Trunk fat mass (kg): 6.5 (5.2–10.7) (at the start) vs. 6.4 (4.7–10.7) (12 months).Giltay et al. (1998) [36]The NetherlandsProspective cohort33 participants(18 trans women; 15 trans men) Hospital SettingAnthropometrics (body mass index; waist-to-hip ratio; total body fat)Not assessedNot assessedMean: trans men = 23 years (range, 16 to 33)trans women = 27 years (range, 18 to 37)Trans menTrans womenEthinyl estradiol plus cyproterone acetate; testosteroneNutrition Status:After 12 months of hormone therapy in the trans men group, the body mass index remained unchanged, while the body fat decreased by 24% and the waist-to-hip ratio increased by 3%. Alternatively, in the trans women group, an increase in all these parameters was observed: 5% in the body mass index, 38% in the total body fat, and 1% in the waist-to-hip ratio.Body mass index: 20.9 ± 2.7 kg/m^2^ (trans women); 21.5 ± 3.0kg/m^2^ (trans men)Total body: 9.5 ± 2.6 kg (trans women); 19.8 ± 4.9 kg (trans men)Mueller et al. (2010) [43]GermanyProspective Study45 participantsHospital SettingAnthropometrics (body mass index; body lean mass and fat mass)Not assessedNot assessedMean:30.4 ± 9.1 yearsTrans menTestosterone undecanoateNutrition Status:Body composition was compared at the start of the testosterone treatment, 12 months, and 24 months afterward. There was a significant increase in lean mass (*p* < 0.01) compared to the body mass index and fat mass.No standardized protocol was used regarding diet or food habits that could influence body composition.Body mass index (kg/m^2^): 24.1 ± 4.5 [at the start] vs. 24.1 ± 4.0 [after 12 months] vs. 24.2 ± 3.8 [after 24 months].Fat mass (kg): 17.8 ± 5.1 [at the start] vs. 17.6 ± 4.4 [after 12 months] vs. 17.5 ± 5.0 [after 24 months]Lean mass (kg): 44.5 ± 6.6 [at the start] vs. 46.3 ± 6.1 [after 12 months] vs. 46.4 ± 5.6 [after 24 months].Cupisti et al. (2010) [32]GermanyProspective Cohort analysis269 participants(29 trans men) University hospital SettingAnthropometrics (body mass index)Not assessedNot assessedMean:29.9 ± 1.1 yearsTrans menTestosterone undecanoateNutrition Status:Body mass index did not show significant changes before or after 1 year of testosterone treatment.Body mass index (kg/m^2^): 23.7 (22.0–25.4) [at the start] vs. 24.2 (22.5–26.0) [after 12 months].Pelusi et al. (2014) [45]ItalyProspective cohort45 participantsHospital settingAnthropometrics (body weight; body mass index; waist and hip circumference, body lean mass, and fat mass)Not assessedNot assessedMean:29.5 ± 1.1 yearsTrans menTestosterone gel; testoviron depot, Testosterone undecanoateNutrition Status:By comparing the effect of three distinct testosterone administrations, the lean body mass increased in all three when comparing post-treatment week 54 with the baseline.Testosterone gel administration:Body weight (kg): 67.3 (59.7–74.9) [baseline] vs. 68.7 (61.5–75.9) [post-treatment week 54] (*p* < 0.0005).Body mass index (kg/m^2^): 23.9 (21.2–26.6) [baseline] vs. 24.3 (21.8–26.9) [post-treatment week 54] (*p* < 0.0005)Waist circumference (cm): 82.7 (73.2–92.1) [baseline] vs. 84.0 (74.9–93.1) [post-treatment week 54] (*p* = 0.256)Hip circumference (cm): 98.7 (91.4–105.9) [baseline] vs. 98.0 (90.2–105.8) [post-treatment week 54](*p* = 0.089)Fat (kg): 21.9 (15.5–28.3) [baseline] vs. 20.4 (16.1–24.7) [post-treatment week 54] (*p* = 0.051)Fat (%):26.7 (19.5–33.9) [baseline] vs. 23.7 (18.6–28.8) [post-treatment week 54] (*p* = 0.001)Testoviron post administration:Body weight (kg): 57.8 (51.2–64.4) [baseline] vs. 61.3 (55.0–67.5) [post-treatment week 54] (*p* < 0.0005).Body mass index (kg/m^2^): 22.3 (19.9–24.6) vs. 23.6 (21.4–25.8) [post-treatment week 54] (*p* < 0.0005)Waist circumference (cm): 73.1 (68.23–78.0) [baseline] vs. 77.5 (72.37–82.2) [post-treatment week 54] (*p* = 0.256).Hip circumference (cm): 94.4 (90.6–98.2) [baseline] vs. 96.0 (91.9–100.1) [post-treatment week 54](*p* = 0.089).Fat (kg): 15.1 (6.6–23.7) [baseline] vs. 14.4 (8.6–20.1) [post-treatment week 54] (*p* = 0.051).Fat (%):26.7 (19.5–33.9) [baseline] vs. 23.7 (18.6–28.8) [post-treatment week 54] (*p* = 0.001).Testosterone undecanoate:Body weight (kg): 59.6 (52.3–66.8) [baseline] vs. 60.5 (53.7–67.4) [post-treatment week 54] (*p* < 0.0005).Body mass index (kg/m^2^): 22.1 (19.5–24.6) [baseline] vs. 22.4 (20.0–24.8) [post-treatment week 54] (*p* < 0.0005).Waist circumference (cm): 81.5 (74.2–88.8) [baseline] vs. 80.6 (73.5–87.7) [post-treatment week 54] (*p* = 0.256).Hip circumference (cm): 97.2 (91.6–102.8) [baseline] vs. 101.8 (95.7–107.9)[post-treatment week 54] (*p* = 0.089).Fat (kg): 15.1 (6.6–23.7) [baseline] vs. 14.4 (8.6–20.1) [post-treatment week 54] (*p* = 0.051)Fat (%):26.7 (19.5–33.9) [baseline] vs. 23.7 (18.6–28.8) [post-treatment week 54] (*p* = 0.001).Deutsch et al. (2015) [33]United States of AmericaProspective cohort57 participants(23 transwomen; 34 transmen)LGBT community health clinic settingAnthropometrics (weight; height; body mass index)Not assessedNot assessedMean: Trans men 29 ± 6.9 years;Trans women 29 ± 9.4 yearsTrans men; trans womenEstrogens (sublingual micronized 17-beta estradiol; transdermal patch; estradiol valerate intramuscular; spironolactone).Testosterone (subcutaneoustestosterone cypionate; testosterone gel; testosterone transdermal patch)Nutritional Status:In the trans men sample, it was observed an increase in the body mass index associated with testosterone therapy. A change in the weight status between the start of the treatment and six months afterward was also observed in the same group. However, correlations regarding these statuses were moderate and weak. No significant changes were detected in the trans women group regarding these parameters.Median weight at the start of the therapy (interquartile range):Trans men: 78 kg (45.4)Trans women: 69.4 kg (17.9)Median weight after six months of therapy(interquartile range):Trans men: 79.4 kg (36.2) (*p* = 0.024)Trans women: 68.8 kg (20.2)Median body mass index at the start of the therapy (interquartile range):Trans men: 29.1 kg/m^2^ (11.2)Trans women: 24.8 kg/m^2^ (4.3)Median body mass index after six months of therapy(interquartile range):Trans men: 30.0 kg/m^2^ (11.4) (*p* = 0.024)Trans women: 23 kg/m^2^ (4.5)Borrás et al. (2021) [30]SpainProspective observational study70 participantsAnthropometrics(weight, height, and body mass index)Not assessedNot assessed18–40 yearsAssigned female at birthTestosterone therapy before gender affirming surgeryNutritional Status:Normal body mass index, mean ± standard deviation = 22.3 ± 2.5 kg/m^2^ (20.0–25.0)Giltay et al. (2004) [37]The NetherlandsRetrospective cohort, observational study81 participantsHospital SettingAnthropometrics (body mass index)Not assessedNot assessedMean:36.7 (21–61) yearsTrans menTestosterone esters; testosterone undecanoateNutrition Status:Body mass index was assessed during the beginning stages of the testosterone therapy, retrospectively, and it increased 3–4 months afterward (*p* < 0.001).Body mass index (kg/m^2^): 22.9 ± 4.5 [at the start] vs. 24.5 ± 3.9 [after 3–4 months].Mueller et al. (2011) [44]GermanyProspective Study84 participantsHospital SettingAnthropometrics (body mass index; body lean mass and fat mass)Not assessedNot assessedMean:36.3 ± 11.3 yearsTrans womenGoserelin acetate; 17-ß oestradiol (estradiol-17 β valerate)Nutrition Status:Body composition was compared at the start of the estrogen treatment, 12 months, and 24 months afterward. The body mass index, fat mass, and lean mass increased as the treatment proceeded due to a shift from lean mass to fat mass.Body mass index (kg/m^2^): 22.3 (21.7–23.0) [at the start] vs. 22.7 (22.0–23.7) [after 12 months] vs. 23.3 (22.3–24.4) [after 24 months] (*p* = 0.001)Fat mass (kg): 10.7 (8.4–14.4) [at the start] vs. 13.1 (9.9–15.6) [after 12 months] vs. 14.3 (10.8–17.2) [after 24 months] (*p* = 0.001)Lean mass (kg): 59.6 (54.6–64.6) [at the start] vs. 57.2 (54.0–64.1) [after 12 months] vs. 55.4 (51.1–58.7) [after 24 months] (*p* = 0.001)Schutte et al. (2022) [48]The NetherlandsProspective observational study95 participants(48 trans women; 47 trans men) Amsterdam University Medical CenterBody mass index (record before the start of the treatment and after 3 and 12 months of treatment)Not assessedNot assessed18–50 years oldTrans women;trans menEstradiol patches plus cyproteroneacetate (CPA);testosterone gelNutritional Status:Baseline body mass index: 23 kg/m^2^ (21–26) in the trans women sample and 23 kg/m^2^ (21–30) in the trans men sample;after three months: 23 kg/m^2^ (21–26) in the trans women sample and 23 kg/m^2^ (22–29) in the trans men sample;After 12 months: 24 kg/m^2^ (22–27) in the trans women sample and 23 kg/m^2^ (22–28) in the trans men sample; adjusting the analyses for change in body mass index did not affect the resultsWierckxet et al. (2014) [52]Norway; BelgiumMulticenter 1-year prospective study as a part of a large prospective study (ENIGI) Clinical Study106 participants(53 trans women; 53 trans men)Hospital settingAnthropometrics (body weight; body mass index; waist and hip circumference, body lean mass, and fat mass)Not assessedNot assessedMean:Trans men 24.6 ± 2.8 yearsTrans women 30.3 ± 4.0 yearsTrans men; trans womenCyproteroneacetate; 17-ß estradiol valerate; testosteroneundecanoateNutritional Status:While the total weight did not suffer any changes during the 12 months of administration in the trans women group (*p* = 0.10 for oral estrogens; *p* = 0.91 transdermal estrogens), it increased significantly in the trans men group (*p* = 0.01), due to an increased in the total lean mass. In trans women, the total body fat mass increased (*p* < 0.001).In the trans women group (oral estrogens):Body mass index (kg/m^2^): 23.1 ± 4.2 (at the start) vs. 23.7 ± 4.4 (12 months) (*p* = 0.42)Waist circumference (cm): 81.2 ± 10.1 (at the start) vs. 79.7 ± 10.5 (12 months) (*p* = 0.21)Hip circumference (cm): 94.2 ± 9.3 (at the start); 98.1 ± 9.3 (12 months) (*p* < 0.001)In the trans women group (transdermalestrogens):Body mass index (kg/m^2^) 26.1 ± 3.5 (at the start) vs. 26.1 ± 3.4 (12 months) (*p* = 0.91)Waist circumference (cm): 91.6 ± 11.1 (at the start) vs. 90.9 ± 10.1 (12 months) (*p* = 0.50)Hip circumference (cm): 96.9 ± 7.8 (at the start); 100.4 ± 7.1 (12 months) (*p* = 0.07)In the trans men group:Body mass index (kg/m^2^): 24.8 ± 5.3 (at the start) vs. 25.6 ± 4.4 (12 months) (*p* = 0.01)Waist circumference (cm): 80.3 ± 13.6 (at the start) vs. 80.1 ± 11.2 (12 months) (*p* = 0.74)Hip circumference (cm): 97.3 ± 10.5 (at the start); 95.4 ± 9.2 (12 months) (*p* = 0.03)Suppakitjanusanta et al. (2020) [49]United States of AmericaRetrospective cohort study.Clinical setting145 participants(105 trans women, 59 entered before hormone therapy; 40 trans men, 25 entered before hormone therapy)Anthropometrics (body weight; height; body mass index) Not assessedNot assessedAge baseline Values:Trans women already on hormone therapy: 43.9 ± 15.6Trans women not on hormone therapy: 32.8 ± 11.7Trans men already on hormone therapy: 40.4 ± 13.1Trans men not on hormone therapy: 24.4 ± 8.8Trans womenTrans menEstradiol (oral; transdermal; intramuscular)SpironolactoneTestosterone (transdermal; intramuscular)Nutritional Status:Body mass index baseline values:Trans women already on hormone therapy: 26.3 ± 4.7 kg/m^2^Trans women not on hormone therapy: 24.7 ± 4.7 kg/m^2^Trans men already on hormone therapy: 26.6 ± 4.2 kg/m^2^Trans men not on hormone therapy: 24.4 ± 5.4 kg/mBody mass index significantly increased in trans womenstarting hormone therapy, on average 0.125 kg/m^2^ (95% CI0.04–0.21, *p*-value = 0.004) per each additional quarter of therapyduration. The same was not observed in trans men after initiation. Body mass index appeared stable following 3 to 6 years of therapy.The study, however, could not correlate hormone therapy to increased body mass index due to potential confounding factors (diet, exercise, mental health status, lifestyles).Vilas et al. (2014) [51]SpainProspective cohort study157 participants (71 trans women; 86 trans men)Hospital settingWeight, body mass index, body fat content, waist and hip circumference169-item quantitative food-frequency; 24h dietary recall; photo-book; energy and nutrients content calculated using DIAL software (available at https://www.alceingenieria.net/nutricion.htm, accessed on 18 May 2024).Not assessedMean = 32.9 ± 9.0 yearsPatients with gender dysphoria (trans women; trans men)Conjugated estrogens orally; estradiol valerate cyproterone acetate testosteroneNutritional Status:The trans women group had a lower BMI than the trans men group, although it was not significant (*p* = 0.71). Around 4.39% of the trans women sample and 6.06% of the trans men group were underweight.Around 12% of trans women sample and 15.15% of trans men were overweight. While more extensive differences were found between the two samples, there were no significant differences between baseline and post-treatment data for body mass index (24.0 ± 5.0 kg/m^2^ vs. 24.1 ± 4.1 kg/m^2^, *p* = 0.12), body fat percentage (27.9 ± 10.7 vs. 28.9 ± 10.2, *p* = 0.20), waist circumference (80.7 ± 10.8 cm vs. 80.0 ± 10 cm, *p* = 0.99) and hip circumference (98.8 ± 10.5 vs. 98.1 ± 9.0, *p* = 0.47). For the body mass index data, no significant differences were observed between the two trans groups (trans woman and trans male) in the sample post-hormone therapy (23.5 ± 3.7 vs. 25.1 ± 4.6, *p* = 0.06)Food Habits/Intake:The sample in the study revealed eating many servings of food and high energy levels (3614 ± 1314 kcal/day).An unbalanced diet with high-fat consumption, especially saturated fats, and cholesterol, was also observed in most of the samples.Together with cross-hormone treatment, the mentioned dietary habits and lifestyle lead to an increase in body fat.Item only includes data during hormonal treatment justifying the body changes; diets observed were hyperlipidemic, hyperproteic, and hypoglucidic.Borger et al. (2024) [29]United States of AmericaCohort166 participants(55 trans women, 111 trans men)Body composition obtained via bioelectrical impedance analysisNot assessedNot assessedMean = 18  ±  1.9 yearsTrans menTrans womenGender-affirming hormone treatment of 1.4 yearsNutrition Status:71% of the cohort sample showed evidence of at least one metabolic syndrome component. Trans men had higher odds of overweight/obesity, elevated/hypertensive BP, elevated triglycerides (TGs), and an atherogenic dyslipidemia index.Chantrapanichkul et al. (2021) [31]United States of AmericaRetrospective cohort study196 participants(134 trans women; 62 trans men)Clinical visitsBody mass indexNot assessedNot assessed<21 years (40.6% trans men; 11.3% trans women)21–34 years (45.2% trans men; 24.2% trans women)≥35 years(24.2% trans men; 42.9% trans women)Trans women;Trans menTestosterone; estradiolNutritional Status:Trans men included a higher proportion of participants with a body mass index of 30 kg/m^2^ or greater than the trans women sample (42% vs. 30%)


### 3.1. Hormone Therapy

Regarding HT, the most frequently used hormones were testosterone, mentioned in sixteen of the studies screened [3,4,5,6,7,8,9,10,11,12,13,14,15,16,17,18,19,20,21,22,23,24,25,26,27,28,29,30,31,32,33,36,37,38,43,45,46,49,50,51,52,53]; estradiol, reported in eleven studies [31,36,38,41,44,46,47,49,51,52,53]; estrogen, used in three [33,46,51]; and also cyproterone acetate, used in three studies [36,47,51]. Other hormones studied include spironolactone [41], anti-androgen [46], and testoviron [28]. In two research papers, no specific HT was detailed [34,42]. In three studies, similarly to the gender identity, broader terms were used, such as “hormone replacement therapy” [39], “sex-steroid hormones” [27], or “gonadotropin-releasing hormone agonist therapy” [35]. Gender-affirming surgery was mentioned in four studies [35,39,40,42].

### 3.2. Nutritional Status

Body mass index (BMI) was the most referenced nutritional status indicator reported in twenty-five of the twenty-seven articles screened [28,30,31,32,33,35,36,37,38,39,40,41,42,43,44,45,46,47,48,49,50,51,52,53]. Lower values of BMI, indicating an underweight status, were found in four studies [35,41,51], two of which were case studies on individuals with a history of eating-disordered behaviors [35,41]. Ewan et al.’s case report had a patient showing a decrease from 26.8 kg/m^2^ to 13.8 kg/m^2^, indicating an undernutrition status [35]. In Maheshwari et al.’s case study, the BMI was only assessed in case 1, revealing that the patient was underweight (17.2 kg/m^2^) [41]. Around 4.3% of the trans women and 6.0% of the trans men samples were underweight in the Vilas et al. prospective cohort study [51].

In four studies, higher values of BMI, indicating overweight and obesity status, were also found [27,40,42,51]. The TM and GNC samples in the analysis by Arikawa et al. both had an average of 27.4 kg/m^2^ [27]. In the Linsenmeyer et al. ten-case study, 70% of the TM sample was obese, with 30% being obese class I, 20% obese class II, and 20% obese class III [40]. In Martison et al.’s sample, the BMI was recorded in two consults, the initial surgical consult, and the most recent subsequent visit at the time of the study [42]. It was observed that 26% of the participants were obese at the initial surgical consult, and 32% were overweight [42]. Referencing Vilas et al.’s cohort study, 12% of the trans women sample and 15.1% of trans men were overweight [51]. A high BMI was also reported to be a risk factor influencing the choice and dosage of HT in Hojbjerg et al.’s [38] questionnaire survey and gender confirmation surgery in Martison et al.’s [42] cross-sectional study.

In addition, in fourteen studies, the comparison of the BMI before and after the HT was reported [28,31,32,33,36,37,43,44,45,48,49,50,51,52], with six studies reporting a change in BMI [28,33,37,44,45,49] and the remaining eight reporting that the BMI did not go through significant changes related to HT [31,32,36,43,48,50,51,52].

Concerning BMI changes, significant changes were observed in BMI after six months of testoviron in Berra et al.’s TM sample [28]. In Guiltay et al.’s retrospective study, BMI was assessed during the beginning stages of the testosterone therapy, retrospectively, and it increased 3–4 months afterward (*p* < 0.001) [37]. Their sample was, similarly, a TM group under testosterone esters and undecanoate HT [37]. The same conclusion was observed in Deutsch et al., associated with testosterone therapies of subcutaneous testosterone, cypionate, gel, and transdermal patch [33]. However, no significant changes were detected in the TW group [33]. Alternatively, BMI was compared at the start of the estrogen treatment, 12 months, and 24 months afterward in Mueller et al.’s cohort [44]. An increase was observed due to a lean-to-fat mass shift [44]. By comparing the effect of three distinct testosterone administrations, lean body bass increased in all three when comparing post-treatment week 54 with the baseline. Lastly, highlighting the Suppakitjanusanta et al. retrospective cohort that compared four groups of trans individuals—two TM groups and two TW groups, one already undergoing HT and the other just starting HT—BMI significantly increased in TW starting hormone therapy [49]. In all groups, BMI appeared stable following 3 to 6 years of therapy [49]. This increase in BMI in TW was also observed in the Giltay et al. sample [36].

Regarding the studies that observed that the BMI did not undergo significant changes, Chantrapanichkul et al. found no differences between BMI values at baseline and after 3 and 12 months of HT [31]. In the Vilas et al. cohort, the research team compared the differences in BMI values of TW and TM groups, with the mean value having more differences between the two groups in the post-treatment data and with higher values reported in the TM group, but still not statistically different [51]. According to the results reported in the longitudinal retrospective cohort article by Schutte et al., the TM sample had a higher frequency of obesity classification (≥30 kg/m^2^). Still, no statistically significant differences were found in BMI when comparing both groups [48]. Similarly, in the cohort analysis of Cupisti et al., BMI did not show significant changes before or after one year of testosterone treatment.

Lastly, comparing high BMI prevalence with non-trans/cis samples, the Sánchez Amador et al. comparative study showed that the TW sample had a higher BMI average than the cis samples [51]. The TW sample was on HT for over six months; Sánchez Amador et al. mentioned that this high prevalence was derived from the body changes from HT [51]. In addition, 71% of Borger et al.’s trans sample cohort showed evidence of at least one metabolic syndrome component, which included obesity/overweight [52].

Weight was the second most referenced parameter, with thirteen research teams reporting [27,28,30,33,35,39,42,45,47,49,51,52]. Similar to the BMI, weight changes were compared between baseline and post-treatment periods [27,28,31,33,42,45,50,51,52]. Arikawa et al. revealed that 66% of the included TM sample had a weight change in the year before the study but did not specify if it was an increase or decrease [27]. Vilas et al. determined a slight rise in the TM group after HT, whereas a slight decrease was observed in the TW group [51]. However, neither change was statistically significant [51]. No changes were noticed in the obesity rate among the Martinson et al. study participants despite self-monitored weight management [42].

Alternatively, Pelusi et al., when comparing post-treatment week 54 with the baseline, detected an increase in body weight, as well as other anthropometric measures [BMI, waist circumference, hip circumference, and body fat] in three distinct testosterone administrations [45]. Deutsch et al.’s research also observed a change in the weight status between the start of the therapy and six months afterward in the TM group [33]. That change was not observed in the TW group [33]. These results were allied with those from Wierckx et al., whose prospective study reported that while the total weight did not suffer any changes during the 12 months of administration in the TW group, it increased significantly in the trans men group [52].

Furthermore, after one year of administration, body weight increased, partly due to an increase in lean body mass, most frequently observed in the TM sample of Van Caenegem et al.’s prospective controlled study [50].

Seven studies reported high waist circumference (WC) [28,36,40,45,50,51,52]. In Linsenmeyer et al.’s research, 60% of the ten cases had a high WC [40]. This parameter had changed when compared at baseline and after HT. However, in two studies, the increase was significant [28,36], and studies reported no significance [45,50,51,52].

Similarly to their increase in BMI, significant increases were observed in waist circumference after six months of testoviron in the Berra et al. study [28]. The same result was found in the Giltay et al. cohort, with the waist-to-hip ratio increasing by 3% in the TM group and by 1% in the TW group [36].

In contrast, in the Pelusi et al. cohort, when comparing three testosterone administrations, WC was one parameter that did not have a statistically significant increase [45]. Van Caenegem et al. reported unchanged waist and hip circumferences after one year of testosterone undecanoate HT [50]. Both Vilas et al. and Wierckx et al. reported a change in the WC of each respective sample. However, it was not significant in either study [51,52].

Another anthropometric measurement observed in the articles screened was body fat (both in percentage and mass), present in nine articles [28,36,40,43,44,45,50,51,52]. High body fat values were obtained by bioelectrical bioimpedance in the Sánchez Amador et al. cross-sectional study [47]. Similarly high values were observed in the Linsenmeyer et al. case study [40] and the Vilas et al. cohort study [51]. Eight studies screened fat mass to determine changes in body composition after HT administration [28,36,43,44,45,50,51,52]. Of these eight studies screened, only one study, Vilas et al., did not show significant differences between baseline and post-treatment body fat percentage [51]. Conversely, significant changes were observed in the fat mass in the Berra et al. study [28]. In the Giltay et al. study, body fat decreased by 24% in the TM sample and increased by 38% in the TW sample [36]. The body fat mass increased as the treatment proceeded in the TW sample in Mueller et al.’s prospective study due to a shift from lean body mass (LBM) to fat mass [44]. Wierckx et al. reported a significant increase in the total body fat in the participating TW [52]. In the TM sample of Van Caenegem et al.’s paper, HT decreased total body fat (9.4%) [50]. In the Pelusi et al. prospective cohort, while there were no statistically significant differences in fat content (kg), the body fat in percentage revealed a significant decrease in all three testosterone treatments [45].

Alongside body fat, LBM was another parameter reported. The results indicated a significant change in LBM, namely, an increase in the TM samples of the Van Caenegem et al. [50], Pelusi et al. [45], and Mueller et al. [43] cohorts, and a decrease in the TW sample of Mueller et al. [44].

### 3.3. Food Habits

Diet quality, intake, and specific patterns were analyzed in three studies [39,40,51]. Vilas et al. used a 169-item quantitative FFQ and a 24h dietary recall and compared the food habits pre- and post-start HT administration [51]. Linsenmeyer W et al. used a three-day food diary in their study, alongside the software ESHA (https://esha.com/products/food-processor/, accessed on 23 September 2024) Food Processor Nutrition Analysis, and the *Dietary Guidelines for Americans 2015–2020*; the acceptable macronutrient distribution ranges for carbohydrates, fat, and protein; and the *Recommended Dietary Allowances* or *Adequate Dietary Intakes* reference were utilized as a base for comparisons with the general population [40]. Kirby and Linde used interviews and surveys to assess qualitative and quantitative data regarding nutrition-related health outcomes, including nutrition knowledge and diet quality [39].

Food habits consisting of low vegetable, grain, and fruit intake and excessive diet in saturated fat due to high consumption of fast-food meals were reported in all three studies [29,37,40].

### 3.4. Eating Disorders

A high prevalence of ED and compensatory behaviors was referred to in six studies [27,34,35,39,40,41].

The methods used in the diagnosis included medical history, in the Maheshwari et al. case report [41], or during a hospital visit, as observed in the Ewan et al. case-study and specific questionnaires. Linsenmeyer et al. used the EAT-26 and ecSI-2 tools to assess the degree of eating competence and the risk of ED [40]. Kirby and Linde (2020) assessed weight loss attempts, weight loss methods, and binge eating tendencies with the questions derived from the University’s College Student Health Survey [39]. The American College Health Association–National College Health Assessment questionnaire was used by Diemer et al. to assess the frequency of the past year’s eating disorder diagnosis, the past month’s diet pill use, the past month’s vomiting and laxative use [34]. Eating Attitudes test score and the Eating Disorder Examination Self-Report questionnaire score were employed by Arikawa et al. as across-sectional analysis of a questionnaire [27].

In the case reports screened, the report performed by Ewan et al. was centered on a patient with anorexia nervosa [35]. The participant also showed compensatory food behavior such as bingeing and purging via vomiting, as well as practicing a diet with restricted energy intake, excessive use of laxatives, and the usage of weight loss supplements [35]. In addition, one of the cases present in Maheshwari et al. had a history of undernutrition due to avoidant/restrictive food intake disorder [41].

Over a third of participants in the Kirby and Linde questionnaire followed a restricted diet to lose weight, 31% engaged in binge eating over the past 12 months, and 50% were attempting to lose weight [39]. In contrast, none of the ten participants screened positive for disordered eating using the EAT-26 tool in the Linsenmeyer et al. case study [40]. According to the ecSI-2.0, two patients scored high in eating competence in Linsenmeyer et al. [40]. In a TM and GNC volunteer sample, 31.4% of respondents exhibited an eating disorder pathology, with 28% reporting that they engaged in eating disorder behaviors, with binge eating being the most frequently observed in Arikawa et al. [27]. Lastly, 15.8% of the transgender sample in the Diemer et al. study was diagnosed with an eating disorder in the past year at the time of the study [34]. Self-reported eating disorder diagnosis and past month’s use of diet pills and vomiting or laxatives were found to be higher among transgender students compared to the non-transgender sample [34].

Another frequent theme was food security, which was present in two studies and often studied in conjunction with eating disorders [27,39]. According to the food security score used in the Arikawa et al. analysis, 54.4% of the TM and GNC samples reported food insecurity. In the twenty-six-college-student Kirby and Linde cross-sectional study sample, over 50% of participants reported eating less due to not having resources [39].
nutrients-16-03280-t002_Table 2Table 2Newcastle Ottawa Quality Assessment Scale scores for the included cohort and case–control studies.Article AuthorSelectionComparabilityOutcomeTotalRepresentativeness of the Exposed CohortSelection of the Non-Exposed CohortAscertainment of ExposureDemonstration That Outcome of Interest Was Not Present at the Start of the StudyAssessment of OutcomeWas Follow-up Long Enough for Outcomes to Occur?Adequacy of Follow-Up of CohortsBerra et al. (2006) [28]-******-6Borrás et al. (2021) [30]-*****-*6Borger et al. (2024) [29]-*******7Chantrapanichkul et al. (2021) [31]-******-6Cupisti et al. (2010) [32]-*******7Deutsch et al. (2015) [33]-*******7Giltay et al. (1998) [36]-*-***-*5Giltay et al. (2004) [37]-******-6Mueller et al. (2010) [43]-******-6Mueller et al. (2011) [44]-*******7Pelusi et al. (2014) [45]-*******7Schutte et al. (2022) [48]-*****-*6Suppakitjanusant et al. (2020) [49]-*****-*7Van Caenegem et al. (2015) [50]-******-6Vilas et al. (2014) [51]-***-***6Wierckx et al. (2014) [52]-******-6Caption: one “*” means one point; “-” means zero points.
nutrients-16-03280-t003_Table 3Table 3Newcastle Ottawa Quality Assessment Scale scores for the included cross-sectional studies.Article AuthorSelectionComparabilityOutcomeTotalRepresentativeness of the SampleSample SizeNon-RespondentsAscertainment of the Exposure (Risk Factor):Assessment of OutcomeStatistical TestArikawa et al. (2021) [27]**-*****7Diemer et al. (2015) [34]-*-***-5Hojbjerg et al. (2021) [38]-*-****5Kirby and Linde (2020) [39]-******6Martinson et al. (2020) [42]---****4Sánchez Amador et al. (2024) [47]-******6Caption: one “*” means one point; “**” means two points; ”-“ means zero points.
nutrients-16-03280-t004_Table 4Table 4Quality scores for the included case studies.Article AuthorResearch Question or Objective Clearly StatedStudy Population SpecifiedConsecutive CasesComparable SubjectsIntervention Clearly DescribedOutcome Measures Clearly DefinedLength of Follow-Up AdequateWell-Described Statistical MethodsResults Well DescribedClassificationEwan et al. (2014) [35]YesYesNANAYesYesNRNRNoFairLinsenmeyer et al. (2020) [40]YesYesNRYesYesYesNRYesYesGoodMaheshwari et al. (2021) [41]YesNoNoYesYesYesNRNRYesFairResmini et al. (2008) [46]YesNoNoYesYesYesNRNRYesFairThe Study Quality Assessment Tool developed by the National Heart, Lung, and Blood Institute was used to assess the quality of these case-studies CD: cannot be determined; NA: not applicable; NR: not reported.

## 4. Discussion

The objective of the present systematic review was to underline the potential differences in the food habits and nutritional status between transgender individuals undergoing HT and those who are not under this therapy, and also what characteristics of the food habits and nutritional status can be observed in the group undergoing HT. This question remains unanswered because only Vilas et al. compared the food habits and nutritional status between HT stages in a prospective cohort study [51]. In the case study developed by Maheshwari et al., before readjusting the HT of one subject who had an eating disorder, the nutritional status and concerns were addressed [41]. However, being a case study, it does not demonstrate the effect [41].

### 4.1. Nutritional Status

The nutritional status of the transgender sample undergoing HT varied between low BMI values (13.8–17.2 kg/m^2^) and high BMI values (25.0–29.9 kg/m^2^). Weight was a nutritional status parameter frequently mentioned in the articles screened. In one cross-sectional study, Arikawa et al. detected a weight change in their sample, though it was not specified how much weight was gained or lost and if it was related to HT [27].

In addition, BMI was considered a general health-related variable in studies where nutritional status was not a primary objective. According to the results reported in the longitudinal retrospective cohort article by Schutte et al., the TM sample presented a higher frequency of obesity (≥30 kg/m^2^). Still, no statistically significant differences were found in the BMI when comparing TW and TM groups before and after HT (*p* = 0.095) [48]. Cupisti et al. also concluded that BMI did not show significant changes before or after one year of testosterone treatment [32]. Moreover, in the study by Muller et al. regarding the effect of testosterone undecanoate on body composition in a TM group, no changes with statistical significance were observed in BMI [43].

However, BMI and weight increased significantly after one year of HT for TW and TM participants in the TM sample of Deutsch et al.’s prospective cohort [33]. An increase was observed in the BMI after six months of testosterone treatment. Conversely, correlations were moderate and weak [33]. The Muller et al. study also showed an increase in BMI after two years of treatment associated with a shift from lean to fat mass in the TW sample [45]. Lastly, changes with statistical significance were observed in the BMI in the samples of Berra et al. [28], Giltay et al. [36], and in the Giltay et al. cohorts [37].

Hojberg et al. argued that presenting a high BMI influenced the choice of starting HT and hormone supplement dosage of HT in TW individuals, and this was the chosen practice in twelve out of the fifteen participating clinics [38]. Other factors, such as severe obesity and diabetes, were also considered essential to assess when administering HT, which added to the importance and influence of nutritional status during HT [38]. However, a high diversity of risk factors was also identified, making it complicated to compare the effect of the different risks and to gather significant data on the safety of feminizing HT [38]. Finally, a high BMI was considered a barrier to gender confirmation surgery, the final procedure related to the gender transition [33].

It should be noted that while BMI is one of the most utilized anthropometric methods of health risk assessment, it has limitations regarding distinguishing between fat and fat-free mass with the potential of healthy individuals being mislabeled as overweight [40].

Other parameters were mentioned. WC was compared regarding gender identity and HT, and no significance between baseline and post-treatment data was found within the cohort of Vilas et al. [51]. An increase in body fat was observed, but similarly to WC, no statistically significant differences were reported between pre- and post-HT [51]. Conversely, in the Berra et al. prospective cohort, significant changes were observed in both the WC and body fat (%) of TM after six months of testoviron [28].

Body fat in kilograms increased significantly after the period of HT in Mueller et al., a prospective study carried out in a TW sample [44]. Total fat mass decreased significantly in the TM sample of Pelusi et al., even with different forms of administration [45]. Lastly, in Giltay et al.’s cohort, after 12 months of HT in their TM group, body fat decreased by 24% [36]. It is important to note that no screenings for diet or food habits were performed in these studies, which could have influenced body composition during HT [43,44].

The results in the present review are in accordance with the bibliography consulted and are the ones available at the time of writing. Klaver et al.’s meta-analysis investigated changes in body weight during HT, and the primary outcomes were centered on total body weight, body fat, and lean body mass [6]. Seventeen of the articles screened were selected, comparable with this review’s twenty-seven. Of those seventeen, ten illustrated the changes in the study’s outcomes, namely, an increase in body fat and a decrease in lean body mass in TW and the inverse in TM [6]. These results were also observed in the current review.

### 4.2. Eating Disorders and Food Insecurity

Restrictive eating behaviors leading to EDs were commonly referred to in the reviewed articles. Two of the case studies included two participants in their respective samples with a history of restrictive food intake disorders, purging, laxative abuse, and weight loss [35,41]. A higher prevalence of these restrictive behaviors was found in the Diemer et al. college sample in trans students compared with non-trans students, showing a higher number of self-reported eating disorder diagnoses [34]. Food insecurity was reported to have a high prevalence, with Arikawa et al. reporting that 54.4% of their sample had experienced it [27]. Cost, access, and food preparation were seen as barriers to including healthy foods in the diet, which is in line with previous reviews. It is also stated that this population’s access to primary health care is often compromised [13,14].

### 4.3. Food Habits

Outside of eating disorders, it was concluded that the transgender population had food habits consisting of low vegetable, grain, and fruit intake and excessive diet in saturated fat as a result of high consumption of fast-food meals and energy-dense and micronutrient-poor foods, which is similar to most of the general non-trans population not meeting the recommendations of those nutrients and food groups [39,40,51]. Regarding micronutrient intake, low potassium and vitamin D levels and high iron and calcium consumption were outlined. These habits were also observed in a previous review by Streed et al. [12].

Differences in the nutritional status and food habits between trans individuals undergoing HT and those who are not undergoing therapy were shown to be low, given that few studies compared these two groups.

In the Vilas et al. longitudinal study, the research team observed modifications in nutritional status due to eating habits and crossed hormone therapy. However, it did not specify how eating habits and HT are connected [51]. Body weight values were compared between the TW and TM groups before and after starting HT, and no significant differences were found [51]. No significant differences between baseline and post-HT were found for BMI, weight, WC, or body fat percentage [51].

The focus on food habits/intake and the potential relation with HT is a major difference between the present review and those referenced. Gomes et al.’s review focused on the extrapolation of the main themes regarding food and nutrition addressed in research on transgender populations [14]. Of the thirty-seven studies selected with good or moderate quality, only five had food and nutrition security as a predominant theme, with another five focusing on nutritional status [14]. No mention of specific food habits or intakes was shown. However, one of the main conclusions of Gomes et al.’s review was that transgender teenagers did have a higher consumption of processed foods compared to non-transgender teenagers [14].

Similarly, in Rozga et al.’s scooping review [13], 10 of the 189 studies examined dietary intake in transgender individuals. This scooping review aimed to illustrate the available bibliography related to nutrition-related intermediate and long-term health outcomes in transgender individuals, dietary intake, the effects of nutrition exposures, and HT [13]. Of the ten articles, six were prospective cohorts, five addressed nutrition-related behaviors, and four mentioned the consumption of specific food groups, such as fruits and vegetables [13]. Regarding the HT effects on nutrition-related intermediate and health outcomes, while a total of 126 studies were screened, food habits or eating intake was not a frequently reported intermediate outcome compared to the anthropometric and body composition measures [13]. These results follow in line with what the present review demonstrated, with BMI being a recurring intermediate outcome and regularly used as a health indicator when comparing post- and pre-HT. They also illustrate the need for more research to be conducted on food habits and eating intakes in the trans adult population, outside of compensatory behaviors related to EDs.

In addition, few studies have used a standardized protocol to analyze diet or food habits. In the present review, three articles that used specific assessment tools were screened [39,40,51]. However, one was a case study on ten TM cases by Linsenmeyer et al. [40], so it did not demonstrate the effect and only focused on one specific gender identity [40]. Another was a cross-sectional survey with a small and homogeneous sample of university students, making it difficult to extrapolate to the rest of the trans population [39]. Lastly, the Vilas et al. cohort was performed with hospital patients with GD. This may lead to Berkson bias, where the selected sample is less healthy and has different exposures than the general population [51]. Consequently, it does not allow for an accurate extrapolation of the general transgender population regarding food habits.

The behaviors observed in all three investigations are modifiable, and by acting early, the risk of future diseases, namely, cardiovascular diseases, may be reduced. It becomes essential to assess a larger sample with people from different age groups and gender identities to better understand the current situation and develop methods to act in this population, providing the best nutritional care and subsequently meeting their nutritional needs. In this aspect, the assessment of dietary intake in future studies using conventional nutritional surveys is of great importance to better understand the association between food and optimal health alongside the gender transition, as well as between risk of illness and health.

### 4.4. Quality Assessment

Given the final scores obtained using the NOS designed for observational studies and the NIH quality assessment tool for case series studies, the studies’ methodological quality ranged from fair to good.

While all studies were of acceptable quality, a high heterogeneity was seen, primarily in the reports of gender identity and the types of HT presented in the articles. Considering this wide variety of HT, assessing potential associations between this wide range of therapies and nutrition status was challenging. A similar difficulty was also observed in a previous work by Klaver et al. [6].

### 4.5. Limitations and Strengths

One limitation of the present review is the small number of studies obtained. Some of the articles screened for this review were case reports and studies with a smaller sample size. Differences were observed in only one study regarding food habits and nutritional status in pre- and post-hormonal treatment [51]. However, this study uses a sample solely categorized with gender dysphoria, which is nowadays not considered the most adequate criteria for classifying members of this target group [51]

Another limitation was the high heterogeneity of different HT regimens and the smaller sample pool that contained both outcomes of interest. This high heterogeneity also extends to the various ways the other outcomes were measured and assessed, examples being the different tools to diagnose ED or the nutritional status, the recruitment for the different samples (both community-based and clinical-based), and a wide variety of HT regimens. Considering this wide variety, assessing potential associations between this wide range of therapies and nutrition status was challenging. This high heterogeneity of assessment methods and samples may explain some of the inconclusive comparison results. A similar difficulty was also observed by Rasmussen et al. [18]. In their review, the lack of consistency in measures used to assess ED outcomes made for more difficult comparisons across the screened studies [18].

Despite these limitations, the main strength of this review is that it relies on an innovative topic. It represents an important step in introducing the potential association between nutritional status and food habits with HT, bringing more attention, and encouraging more investigations on the topic. In addition, this review summarizes the current available scientific knowledge on the association between HT, nutritional status, and food habits.

### 4.6. Future Research

Transgender health is a new area in the field of nutrition, given the unique gender experience of this population, which may not follow the standard dietary and nutritional recommendations, as well as the specific nutritional concerns related to HT. Since this is a relatively new target group, few clinical practice guidelines and information are available.

Future investigations should be conducted, and the results and data presented here should be used as a basis. Further associations between nutritional status, HT, food and dietary habits, and ED should be studied with more extensive samples for more substantial statistical power. In addition, future studies should continue to utilize different nutritional parameters alongside BMI and weight to gain a clearer view of the nutritional status and its evolution alongside the gender transition process.

## 5. Conclusions

Transgender nutrition is an area with very few studies about nutritional status, food insecurity, food habits, and their association with HT, with a limited level of knowledge and skills regarding transgender health among general health professionals.

Nutritional status is perceived as a relevant factor when administering HT and seems to be influenced by this therapy. The transgender samples undergoing HT considered in this review showed high BMI, alongside high WC and high body fat, indicating overweight status. In other cases, lower to medium BMI values were also reported, partly related to restrictive eating behaviors. Data were shown to be inconclusive when comparing HT-undergoing participants and non-HT-undergoing participants. As more studies regarding transgender health are being undertaken, future research should aim to assess the nutritional status and specific food habits that are influenced by HT.

The results of this current review highlight the need to give more attention to food habits and nutritional intake as a part of the nutritional status assessment in the transgender population until the release of new specific guidelines for this target group and results from prospective cohort studies that focus on these effects.

## Figures and Tables

**Figure 1 nutrients-16-03280-f001:**
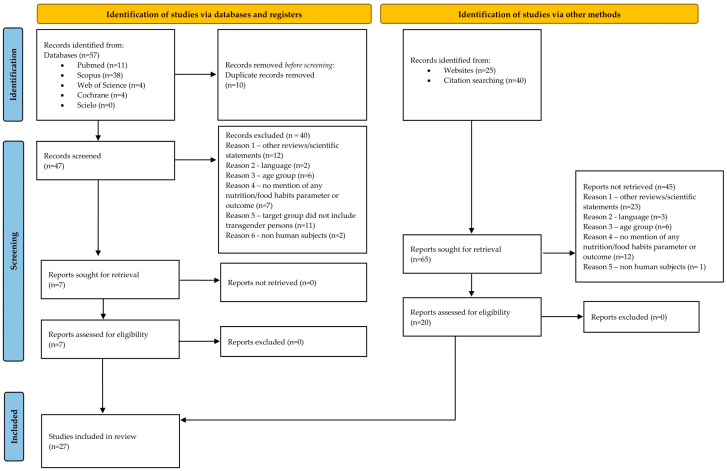
PRISMA 2020 flow diagram.

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
