# Peer review of "Relationship between Food Habits, Nutritional Status, and Hormone Therapy among Transgender Adults: A Systematic Review"

_nutrients, 2024, doi:10.3390/nu16193280_

Round 1

Reviewer 1 Report

Comments and Suggestions for Authors

The authors have undertaken a fascinating and timely topic, one that is increasingly relevant and needs to be addressed from multiple perspectives. The importance of systematically reviewing the nutritional status, eating habits, and related health issues among transgender individuals cannot be overstated, and it is highly commendable that the authors have taken on this essential task. This systematic review is a significant contribution to the field, and your efforts are greatly appreciated.

1. Introduction:

The authors present a well-rounded and insightful introduction, highlighting the unique nutritional challenges faced by the transgender population, particularly in the context of hormone therapy. The discussion is thorough, covering the physiological impacts of HT on nutritional status, including weight changes, body composition, and associated risks such as cardiovascular diseases. The introduction also underscores the gap in existing nutritional guidelines tailored to transgender individuals, making a strong case for the need for further research.

Major revisions:

While the introduction is comprehensive, it would benefit from a more balanced focus by including the nutritional status of trans men, as the current discussion centers predominantly on trans women. Specifically, it would be worthwhile to explore the potential for increased risk of eating disorders in trans men, possibly influenced by factors such as the presence of breasts in those who have not undergone or do not wish to undergo mastectomy, or due to their overall body morphology. Discussing these aspects could shed light on the psychological and physical stressors unique to trans men and would contribute to a more holistic understanding of the nutritional challenges faced by the entire transgender population.

2. Material and methods:

Major revisions:
The authors have done well to include a comprehensive description of the selection process within the "Materials and Methods" section, providing clarity on how studies were identified and screened. However, there are a few key areas that require further attention. While the authors describe the process thoroughly, they should also mention that articles were obtained from other sources beyond the primary databases, and these sources should be clearly acknowledged. Additionally, it is important to specify that one of the phases involved a full-text review of the remaining studies after the initial title and abstract screening. It is recommended that Figure 1, which illustrates the flow of study selection, be moved to the "Results" section, as advised by PRISMA guidelines. Furthermore, the figure should be updated to include the reasons for excluding the 41 articles identified through other sources. These adjustments will align the manuscript more closely with best practices for systematic reviews, enhancing both the clarity and transparency of the study.

3. Results:
Major revisions:

The results section of the study presents a comprehensive analysis of the selected articles, covering various aspects such as nutritional status, hormone therapy effects, eating habits, and the prevalence of eating disorders among transgender individuals. The section provides detailed information on the distribution of participants across different studies, the types of hormone therapies used, and the changes in body composition and nutritional indicators, such as BMI, weight, and fat mass. The review also discusses the quality of diet and the high prevalence of eating disorders observed in several studies. However, there are some areas that could be improved for clarity and completeness:

Citation of Articles: When mentioning that specific parameters, such as BMI or eating disorders, were evaluated by multiple articles, the authors should cite the relevant studies directly in the text. For instance, if six articles assessed a particular parameter, each should be cited to provide readers with clear references.

Enhancements to the Tables: The tables should include an additional column dedicated to eating disorders, specifying which disorder was observed if present. Additionally, another column should be added to indicate the method used to determine the prevalence of the eating disorder, such as specific questionnaires or clinical assessments.

Inclusion Criteria Concerns: There are several articles included in the review that feature age ranges which do not seem to align with the stated inclusion criteria of participants being 18 years or older. For example:

    • Furutani et al. reported a range of 17 to 71 years.
    • Ávila et al. included participants aged 13 to 22.
    • Linsenmeyer et al. included participants aged 12 to 23.
    • Klaver et al. also included minors.
    • Sánchez-Toscano et al. had a range of 16 to 22 years.
    • VanKim et al. included participants aged 10 to 23 years.

These age ranges suggest the inclusion of minors, which does not align with the inclusion criteria mentioned in both the article and the registered protocol on PROSPERO. As these criteria are formalized in the PROSPERO registration, they must be strictly adhered to. Consequently, these articles should be excluded from the review, and their exclusion should be clearly indicated in the flow diagram.

Risk of Bias Description: While the risk of bias has been assessed using the Newcastle Ottawa Scale and the NIH tool, there should be a clear description in the text of how the risk of bias was evaluated for the studies presented in Tables 2, 3, and 4. This description should include a discussion of the results of these assessments, helping readers to understand the potential limitations of the included studies and the overall strength of the evidence presented.

4. Discussion:
The discussion section as currently presented is notably incomplete and lacks several critical components that are essential for a comprehensive and impactful discussion. The authors have touched upon the key findings of their study, such as the differences in nutritional status and eating behaviors among transgender individuals undergoing hormone therapy (HT), but these points need to be further developed to provide a more comprehensive analysis.

To begin with, the discussion should start with a stronger summary statement that highlights the significance of the study’s findings within the broader context of transgender health and nutrition. While the authors have mentioned some of the main outcomes, a more explicit statement on the importance of these findings for the field would help to underline the study’s contribution.

Moreover, while the authors compare some of their findings with previous studies, this comparison needs to be more thorough. The discussion should explicitly position the current study in relation to existing research, emphasizing how these findings confirm, extend, or challenge prior knowledge. For instance, the inconsistencies in BMI changes reported in different studies could be explored more deeply, considering factors such as the duration of HT or different hormonal regimens. This would not only contextualize the results but also demonstrate the study's relevance and novelty.

The broader scientific implications of the findings are briefly mentioned, but this section should be expanded. The authors should discuss in more detail how their results might influence clinical practices, particularly in the development of nutritional guidelines tailored to transgender individuals. Additionally, the potential impact on public health policies could be explored, providing a more comprehensive view of how these findings could be applied in real-world settings.

Furthermore, the discussion would benefit from a forward-looking perspective, outlining how these findings can guide future research. The authors should consider suggesting specific areas where further studies are needed, such as the long-term effects of HT on nutritional status or the psychological factors contributing to eating disorders in transgender populations. This would help to frame the study as part of an ongoing research dialogue and encourage further investigation in this underexplored area.

The methodologies used in the study, particularly any novel approaches, should also be highlighted more prominently in the discussion. If there were any unique aspects of the study design or data collection that could serve as a model for future research, these should be emphasized. This not only adds value to the current study but also provides useful insights for other researchers working in the field.

Finally, while the authors mention the prevalence of eating disorders among transgender individuals, this point could be strengthened by including a more detailed analysis of the methods used to assess these disorders. The discussion should also consider the limitations of the current study, particularly regarding the inclusion criteria and the potential bias introduced by the age range of participants in some of the studies reviewed. Addressing these limitations transparently would enhance the credibility of the findings and provide a clearer basis for the conclusions drawn.

5. Additionally, I noticed that Figure 1 has been included in the supplementary materials, despite already being presented in the main text. Could you clarify the rationale behind this decision? Typically, figures that are essential to understanding the study are included only in the main manuscript to avoid redundancy.

As a suggestion for future systematic reviews, it is generally recommended to involve an additional author in the search and selection process. This third author can help ensure impartiality, particularly in cases where there is disagreement between the two primary reviewers regarding the selection criteria. Including a third reviewer can enhance the reliability and objectivity of the review process.

Author Response

The authors acknowledge this important appreciation of the manuscript and express recognition for the pleasant and very useful comments that led to an improvement of the manuscript. All efforts were made to respond to the important remarks and incorporate all the suggestions in this revised manuscript.

Reviewer #1

  1. Introduction:

The introduction was completed to include information concerning nutritional status and increased risk of eating disorders in trans men, as suggested.

  1. Material and methods:

Articles that were obtained from sources other than the primary databases and these sources were acknowledged in the Prisma Flow Diagram.

The information that a full-text review of the remaining studies took place after the initial title and abstract screening was added to this section.

The Prisma Flow Diagram was moved to the "Results" section, and this diagram was updated to include the reasons for excluding the (now) 40 articles identified through other sources.

An updated version of the visual abstract was uploaded.

  1. Results:

The relevant studies were directly cited in the text, as indicated.

An additional column in Table 1 was dedicated to eating disorders, specifying which disorder was observed if present, as well as the method used to determine the prevalence of the eating disorder.

Regarding the “Inclusion Criteria Concerns: There are several articles included in the review that feature age ranges that do not seem to align with the stated inclusion criteria of participants being 18 years or older.” these six studies indicated by the Reviewer were removed from the present study.

Concerning the Risk of Bias Description, a description of how the risk of bias was evaluated for the studies presented in Tables 2, 3, and 4 was inserted, as well as a discussion of the results of these assessments. 

  1. Discussion:

The discussion section was extensively edited, deepened, and completed. A summary statement at its beginning and a final statement on the importance of these findings for the field were inserted.

The section about the scientific implications of the findings was expanded, as recommended.

The approaches used in the study that conferred novelty to this review were mentioned more objectively.

We inserted a more detailed analysis of the methods used to assess the prevalence of eating disorders. Limitations of the present study were edited to include information on the inclusion criteria and the possible bias caused by the age range of participants in several studies.  

  1. Figure 1:

We have also introduced Figure 1 in supplementary material because we were unsure about our success in presenting it in the manuscript template.

Reviewer 2 Report

Comments and Suggestions for Authors

Ivo Sousa and Teresa F. Amaral submitted to Nutrients a systematic review, focusing on the relationship between food habits, nutritional status and HT among TGAs.

This recent review appears well structured, however it requires some essential clarifications.

The citation n. [10] is not indicated in the text.

In the text, at line 57 which refers to citation n. [11], the indicated citation is not by Klaver (as stated in the paper).

At line 162 it is written: “(please insert Table 1)”. The same for line n. 244: “(please insert Table 2-4)”. What does this mean exactly? It seems to me that the tables are already included.

In paragraph 4.5 all the limitations of this review have to be indicated, not only the main one (low number of studies on this topic).

This manuscript should be integrated, with adequate critical spirit, with the take-home messages that emerge from the study, with the practical implications related to the topic under investigation. Basically, how is this review innovative? What does it bring that is new and certainly shareable with the scientific population, compared to what is already sufficiently known on this topic?

Comments on the Quality of English Language

Minor editing of English language required.

Author Response

The authors acknowledge this important appreciation of the manuscript and express recognition for the pleasant and very useful comments that led to an improvement of the manuscript. All efforts were made to respond to the important remarks and incorporate all the suggestions in this revised manuscript.

Reviewer #2

All the citations were verified and edited.

We had some difficulties including the tables in the template and left a comment to the editing team. These comments were deleted.

The limitations were completed, and considerations about this review's practical implications and innovative aspects were inserted at the end of the Discussion section.

The English language was edited.